# Methane emissions from Arctic landscapes during 2000-2015: An analysis with land and lake biogeochemistry models

Xiangyu Liu[1], Qianlai Zhuang[1,2,3]

[1]Department of Earth, Atmospheric, Planetary Sciences, Purdue University, West Lafayette, IN, USA
[2]Department of Agronomy, Purdue University, West Lafayette, IN, USA
[3]Purdue Climate Change Research Center, West Lafayette, IN, USA

*Correspondence to*: Qianlai Zhuang (qzhuang@purdue.edu)

**Abstract.** Wetlands and freshwater bodies (mainly lakes) are the largest natural source of greenhouse gas $CH_4$ to the atmosphere. Great efforts have been made to quantify these source emissions and their uncertainties. Previous research suggests that there might be significant uncertainties coming from "double accounting" emissions from freshwater bodies and wetlands. Here we quantify the methane emissions from both land and freshwater bodies in the pan-Arctic with two process-based biogeochemistry models by minimizing the double accounting at the landscape scale. Two non-overlapping dynamic areal change datasets are used to drive the models. We estimate that the total methane emissions from pan-Arctic are $36.46 \pm 1.02$ Tg $CH_4$ yr$^{-1}$ during 2000-2015, of which wetlands and freshwater bodies are $21.69 \pm 0.59$ $CH_4$ Tg yr$^{-1}$ and $14.76 \pm 0.44$ Tg $CH_4$ yr$^{-1}$, respectively. Our estimation narrows the difference between previous 'bottom-up' ($53.9$ Tg $CH_4$ yr$^{-1}$) and top-down ($29$ Tg $CH_4$ yr$^{-1}$) estimates. Our correlation analysis shows that air temperature is the most important driver for methane emission of inland water system. Wetland emissions are also significantly affected by vapor pressure while lake emissions are more influenced by precipitation and landscape areal changes. Sensitivity tests indicate the pan-Arctic lake $CH_4$ emissions were highly influenced by air temperature, but less by lake sediment carbon increase.

## 1. Introduction

Atmospheric methane ($CH_4$) is one of the major greenhouse gasses which contributes to about 20% of the warming effect, second only to carbon dioxide ($CO_2$). Atmospheric methane concentrations have risen 2.5 times since the beginning of the industrial age (Hamdan and Wickland, 2016). However, its 100-year global warming potential is around 28 times higher than $CO_2$ (27.2 in non-fossil origin and 29.8

in fossil origin; IPCC, 2021). Previous studies have suggested that inland water systems (wetlands and freshwater bodies) are the single largest natural source of the greenhouse gas $CH_4$ (Saunois et al., 2020), both of which have been found to increase under changing climate. Wetland $CH_4$ emissions are the largest natural source in the global $CH_4$ budget, contributing to 60–80% of natural $CH_4$ emissions, equivalent to roughly one-third of total natural and anthropogenic emissions (Quiquet et al., 2015; Hopcroft et al., 2017). Under the RCP 2.6 scenario, climate change-induced increases in boreal wetland extent and temperature-driven increases in tropical $CH_4$ emissions will dominate anthropogenic $CH_4$ emissions by 38 to 56% toward the end of the 21st century (Zhang et al., 2017).

Likewise, lakes are the second largest $CH_4$ source of all inland water emissions after wetlands (Kyzivat, et al., 2022), accounting for approximately 30% of biogenic methane emissions (Guo et al., 2020). They are especially common in high latitudes and account for about 10% of the boreal landscape (Guo et al., 2020). This high coverage of lakes especially the extensive shallow seasonally ice-covered ones in the subarctic landscapes has been considered as a major source of atmospheric methane in northern high latitudes (Bastviken et al., 2011, West, et al., 2016). Unlike wetlands, shallow lakes have the highest methane emission potential in the cold season which dominate the spring methane release in the pan-Arctic area (Jammet et al., 2015), since the ice layer in winter prevents methane from being oxidized by the atmospheric oxygen and from being released to the atmosphere, methane accumulated during the winter can be released in a large pulse during the spring ice melt (Phelps et al., 1998; Guo et al., 2020). In addition, due to the considerable total lake area and the substantial shallow lakes in the area of 40-70° N, this region was also found to be the dominant contributor (~30%) of global lake diffusive $CH_4$ emissions (Li et al., 2020). However, in comparison with land methane emission studies, less work has been done on studying lake $CH_4$ emissions through process-based modeling (Saunois et al., 2020), especially for the pan-Arctic region.

To date, although great efforts have been made to quantifying the uncertainties of global wetland and lake methane emissions separately (Liu et al., 2020; Guo et al., 2021), there are still significant differences between the estimates of the Arctic $CH_4$ natural sources using 'bottom-up' method which aggregated lakes, wetlands and coastal waters as $CH_4$ sources (32–112 Tg $CH_4$ $yr^{-1}$; McGuire et al., 2009; Saunois et al., 2020) and 'top-down' method which determines the emissions based on the spatial and temporal variability of atmospheric $CH_4$ concentration measurements (15–50 Tg $CH_4$ $yr^{-1}$; AMAP, 2015). In those studies, there are potential "double accounting" issues for certain areas of wetlands and lakes

using low-resolution wetland and lake distribution data (Thornton et al., 2016). Specifically, some small lakes and ponds might have been considered as lakes using lake models while wetland modeling might have also treated those as wetlands, therefore being accounted for twice in the regional methane emission estimation.

Here we use two process-based biogeochemical models, the Terrestrial Ecosystem Model (TEM-MDM, Liu et al, 2020) and the Arctic Lake Biogeochemistry Model (ALBM, Guo et al., 2020), along with two dynamic area datasets for both wetlands (WAD2M, Version 2.0; Zhang et al., 2022) and lakes (GLCP; Meyer et al., 2020) ecosystems which cover the inland water systems throughout the landscape without overlap, to quantify the methane emissions considering the impact of the landscape changes in both land ecosystems and freshwater bodies in the study region for the period 2000-2015.

## 2.  Method

### 2.1.  Model description

The Terrestrial Ecosystem Model (TEM) is a process-based biogeochemistry model which considers carbon, nitrogen, water, and heat processes in terrestrial ecosystems and was originally used to simulate ecosystem carbon and nitrogen dynamics (Melillo et al., 1993; Zhuang et al., 2001, 2002, 2003, 2004, 2007, 2013). The model considers important freeze-thaw processes and explicitly integrates soil thermodynamics in permafrost and non-permafrost region biogeochemical processes. It is also coupled with a complex hydrological module that enables the modeling of soil moisture profiles and water table depths in upland and wetland ecosystems. Zhuang et al. (2004) also developed a Methane Dynamics Module (MDM), which was integrated into TEM to estimate $CH_4$ emissions from northern high-latitude regions and further revised and extrapolated to the global scale to quantify soil methane consumption (Zhuang et al., 2013). Recently, Liu et al. (2020) revised the model to the version we used in this study by taking into account several more detailed land methane cycling processes, including various types of wetlands in different regions based on plant functional types, the impact of above-soil surface water on methane transport, and cumulative vertical methane concentrations in soil, such that it can give a more precise methane estimate on the global scale.

The Arctic Lake Biogeochemistry Model (ALBM) is a 1-D process-based climate-sensitive lake biogeochemical model originally developed for simulating $CH_4$ production, oxidation, and emission in

Arctic lakes (Tan et al., 2015; Tan and Zhuang 2015a, 2015b) and later revised to predict both thermal and carbon dynamics of aquatic ecosystems in boreal lakes (Tan et al., 2017; Guo et al., 2020), and it was then successfully applied to temperate lakes (Tan et al., 2018; Guseva et al., 2020). Recently, the

ALBM is also shown to be capable of simulating global lake thermal dynamics (Guo et al., 2021). The model consists of several modules, including those for the water/sediment thermal circulation, conceptualized as the water thermal module (WTM) and the sediment thermal module (STM), and those for the gas diffusive and ebullition transportation, conceptualized as the bubble transport module (BTM) and the dissolved gas transport module (GTM) (Tan et al., 2015). The model also covers the radiative

transfer processes and the water/sediment biogeochemistry, including the terrestrial ecosystems' organic carbon loading, the microbial and photochemical organic carbon degradation, the photosynthesis for inorganic carbon fixation, and phytoplankton biomass loss through respiration for further simulation of $CO_2$ dynamics. The ability of ALBM to simulate and represent the thawing and freezing cycles of sediments in thermokarst lakes and the organic carbon inputs induced by thermokarst activities, the

degradation of dissolved organic carbon through photochemical mineralization, and the mobilization and mineralization of labile organic carbon in the deep sediments of yedoma lakes is crucial for understanding the carbon dynamics in Arctic lakes which makes it a better choice for simulating Arctic lake methane emission than other lake models that are usually lacking these processes (Tan et al., 2017).

### 2.2. Input Data

Here we use two global dynamic area changing datasets for both wetland and lake ecosystems. For wetlands, the Wetland Area and Dynamics for Methane Modeling (WAD2M) Version 2.0 was used as the TEM-MDM model input as transient wetland inundation fraction data. The dataset following the same processing method as Version 1.0 (Zhang et al., 2021), which was used for quantifying the global methane budget for 2000-2017 (Saunois et al., 2020), but included a few updates on the static inventories applied

in WAD2M and used the same monthly SWAMPS version 3.2 (Jensen and McDonald, 2019), was provided for the Global Carbon Project wetland $CH_4$ (GCP-$CH_4$) model intercomparison. Compared to the previous one, the new version applied Global River Width from Landsat (GRWL) Database (https://zenodo.org/record/1297434; Allen and Pavelsky, 2018) and HydroLAKES (https://www.hydrosheds.org/images/inpages/HydroLAKES_TechDoc_v10.pdf; Messager et al., 2016)

instead of the Joint Research Center Global Surface Water (GSW) dataset (Pekel et al., 2016) to remove

inland freshwater systems, defined as lakes, ponds, and rivers and the time period was extended to 2000-2020. Land cover data, which are used to assign parameters to each grid cell, followed Liu et al. (2020), from which vegetation type distribution is from Melillo et al. (1993) and soil texture is from Zhuang et al. (2003).

For lake simulation, we used the Global Lake area, Climate, and Population dataset (GLCP; Meyer et al., 2020) as the dynamic input for ALBM model. Using the HydroLAKES database version 1.0 for the locations and numbers of lakes, the GLCP contains over 1.4 million lakes of at least 10 ha in surface area, with annual surface area (identified as permanent or seasonal water) from 1995 to 2015, paired with annual basin-level temperature, precipitation, and population values HydroLAKES is a global database

of all lakes with a surface area of at least 10 ha based on inventories using geo-statistical approaches. Since GLCP directly uses HydroLAKES to determine the lake locations and numbers, and HydroLAKES is also the dataset WAD2M used to remove inland freshwater bodies, thus, the combination of these two datasets (GLCP and WAD2M 2.0) covers the inland water systems throughout the landscape and will not overlap with each other. Hence using these two dynamic datasets will minimize "double accounting"

problem, which refers to some lakes and ponds being accounted for twice in both regional lake and wetland methane emission estimation at the landscape scale (Thornton et al., 2016). We further classified the lakes into four types based on their location and permafrost thawing type in the pan-Arctic area (above 45° north), including yedoma thermokarst lakes (yedoma/YDM), non-yedoma thermokarst lakes (thermokarst/TMK), non-thermo boreal lakes (boreal/BRL), and temperate lakes (temperate/TMP). From

which, yedoma and thermokarst lakes are classified based on circum-polar Yedoma map (Jens et al., 2022) and Arctic Circumpolar Distribution and Soil Carbon of Thermokarst Landscapes (Olefeldt et al., 2016), non-thermo boreal lakes and temperate lakes were defined on whether their location is above 60° north. At the end, there are total 1,248,478 lakes were simulated, including 101,852 yedoma lakes, 249,434 non-yedoma-thermokarst lakes, 390,687 non-thermo-boreal lakes, and 506,505 temperate lakes. Because

the time period is different for these two datasets (2000-2020 for WAD2M and 1995-2015 for GLCP), we chose the overlap years 2000-2015 as our simulation time period.

        For the climate forcing data, we used GSWP3-W5E5 and 20CRv3-ERA5 datasets (gswp3-w5e5_obsclim_global_daily         and         20crv3-era5_obsclim_hurs_global_daily         , https://data.isimip.org/10.48364/ISIMIP.982724; Lange et al., 2022), both are factual climate input daily

dataset with a resolution of 0.5° × 0.5° globally provided by the Inter-Sectoral Impact Model

Intercomparison Project (ISIMIP). These forcing data were used for both models to ensure that no additional uncertainties are introduced. Air temperature, surface pressure, wind speed at 10m, relative humidity, precipitation, snowfall, downward short-wave radiation and downward long-wave radiation were used in ALBM model as input forcing. For TEM-MDM model simulation, we only used air temperature, relative humidity, precipitation, and downward short-wave radiation, where air temperature and relative humidity were used to calculate the vapor pressure as another input.

## 2.3.  Model parameters

The model parameters are derived from previous studies, both of which did the parameter calibration and validation on a global scale (Liu et al., 2020; Guo, et al., 2021). For TEM-MDM, 15 key parameters involved in wetland methane oxidation and production processes were calibrated and validated at the site level (15 sites for calibration and 14 sites for validation) using the Shuffled Complex Evolution Approach (SCE-UA). Other information, such as vegetation type, soil texture, and wetland type, were also set based on site observations.  For ALBM, 58 freshwater lakes of varying shapes, locations, climates, and landscapes were used for the calibration of nine lake sediment property related parameters. The calibration process used the Sobol sequence sampling method to generate a perturbed parameter ensemble (PPE) of 10,000 samples from the parameters space and then the Monte Carlo method was applied to simulate this PPE for each lake. Six years of the observation data from each lake were used for calibration and the rest were used for validation.

## 2.4.  Simulation protocols

Model simulations followed different protocols for different models. In wetland simulation (using TEM-MDM), the Terrestrial Ecosystem Model 5.0 (TEM5) was first run in the same simulation area and time period to get the net primary production (NPP) and leaf area index (LAI), the outputs were then fed to TEM-MDM as input to calculate methane emissions. For TEM5 simulation, we first did the spin-up run 10 times with 40 years per spin before the transient simulation to let the model reach a steady state using the first 40-year (1901-1940) input data, 120 years (1901-2020) transient simulation was run in TEM-MDM while the first 100 years simulation was used as spin up. For lake simulation using ALBM, as discussed in section 2.2, the lakes were classified into four types based on their location and permafrost thawing type. We further grouped each type of lakes based on their surface area (<1 km$^2$, 1-10 km$^2$, >10

km$^2$) and depth (< 3 m, > 3 m) and whether they are in the same 0.5° × 0.5° pixel so that lakes in the same groups will be driven by the same meteorology input data. Different types of lakes used different parameter sets derived from calibration. For all the simulations, a spin-up period of 10 years was run first.

## 2.5. Sensitivity test

Sensitivity tests were conducted towards lake emissions simulation in three aspects. According to the previous studies, under the SSP5-8.5 scenario, the temperature will increase roughly by 4-6 °C (IPCC, 2021; Huang et al., 2022) and the precipitation exhibits an increasing trend at a rate of 10.28 mm/decade in the northern hemisphere, corresponding to ~13-18% increase by the end of the 21[st] century (Chen et al., 2014; Du et al., 2022). Therefore, we rerun the simulation by 1) increasing the daily temperature by 5 °C; 2) increasing the daily precipitation by 15%, where both rain and snowfall were considered; and 3) adding additional 15% carbon into lake sediments to simulate the influence of permafrost thawing due to global warming. For temperature and precipitation, we directly modified them at the data input step. For lake sediment carbon, we assumed that the additional carbon transferred straightly from old organic matter in thawing permafrost (old organic carbon pool) to new organic matter at the water-sediment interface (young organic carbon pool) and changed it by altering the labile carbon density ($C_{labile}$) (Tan et al., 2015). Because the old organic carbon pool may only contribute to $CH_4$ production in the permafrost thaw bulb under yedoma and thermokarst lakes, we just altered the corresponding $C_{labile}$.

## 3. Results

### 3.1. Temporal dynamics of methane emissions at the landscape scale

The ALBM model simulation driven with the GLCP dataset indicates that the methane emission from lakes in the pan-Arctic region ranges from $11.88 \pm 0.18$ Tg $CH_4$ yr$^{-1}$ in the year 2000 to $18.20 \pm 0.31$ Tg $CH_4$ yr$^{-1}$ in the year 2015 with a mean value of $14.76 \pm 0.44$ Tg $CH_4$ yr$^{-1}$. For different types of lake, we estimate $6.41 \pm 0.05$ Tg $CH_4$ yr$^{-1}$ for temperate lakes, $3.07 \pm 0.09$ Tg $CH_4$ yr$^{-1}$ for boreal lakes, $2.36 \pm 0.28$ Tg $CH_4$ yr$^{-1}$ for thermokarst lakes, and $2.92 \pm 0.07$ Tg $CH_4$ yr$^{-1}$ for yedoma lakes, respectively. The TEM-MDM model driven with WAD2M 2.0 inundation data estimates land ecosystem net emissions of $21.69 \pm 0.59$ Tg $CH_4$ yr$^{-1}$, ranging from $19.44 \pm 0.63$ in 2009 to $23.87 \pm 0.76$ in 2007. Combined the two model simulations along with two dynamic area change datasets, we estimate that the total annual methane emission from inland water systems in the region of 45° N north during 2000-2015 is $36.46 \pm$

1.02 Tg CH$_4$ yr$^{-1}$, with the lowest value of 31.91 ± 0.61 Tg CH$_4$ yr$^{-1}$ in the year 2000 and the highest value of 41.09 ± 1.35 Tg CH$_4$ yr$^{-1}$ in 2015 (Fig. 1a).

Fig. 1b shows the landscape change over the 2000-2015 period. From which, the wetland area was calculated using inundation fraction data and the lake area was directly derived from the GLCP dataset. The total annual average area of the inland water system in the study region is 3,090,690 ± 38,203 km² (mean ± standard deviation) with a minimum value of 3,039,565 km² in 2003 and a maximum of 3,169,494 km² in 2015. The total wetland area is 1,122,493 ± 36,303 km² ranging from 1,074,079 km² (2009) to 1,199,428 km$^2$ (2010). For lakes, the total area ranges from 1,919,652 km² in 2003 to 1,996,625 km² (1,968,197 ± 19,708 km²).

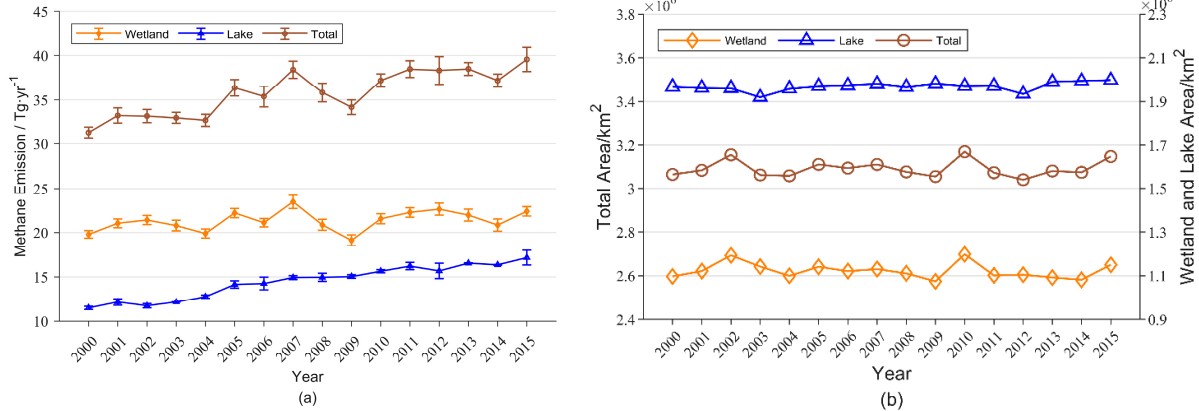

**Figure 1. Annual (a) methane emissions and (b) landscape change.**

### 3.2. Spatial variations of landscape-level methane emissions

Spatial wetland and lake methane emissions are shown in Fig. 2a and b separately. West Siberia Lowland and the Hudson Bay Lowland were the two strong sources. There are many sporadic high emission sources in wet tundra and small wetlands in boreal forest regions, and river and coastal floodplains. Although a majority of lakes are located in the northern Hudson Bay area, they all have low emissions at around 1 g CH$_4$ m$^{-2}$ yr$^{-1}$, compared to which, lakes near Mackenzie River delta of Canada and the Hudson Bay Lowland area have a relatively higher emission at 50 g CH$_4$ m$^{-2}$ yr$^{-1}$, as well as lakes in northern Europe such as Sweden, Finland, and the northwest corner of Russia (around Lake Onega). Fig. 2c shows the methane emission for inland water systems in the pan-Arctic area, it is worth noting

that the average emissions of the lake are usually higher than the emission of the wetlands around the lake, indicating that lakes emit more methane than wetlands in same the region under the same conditions.

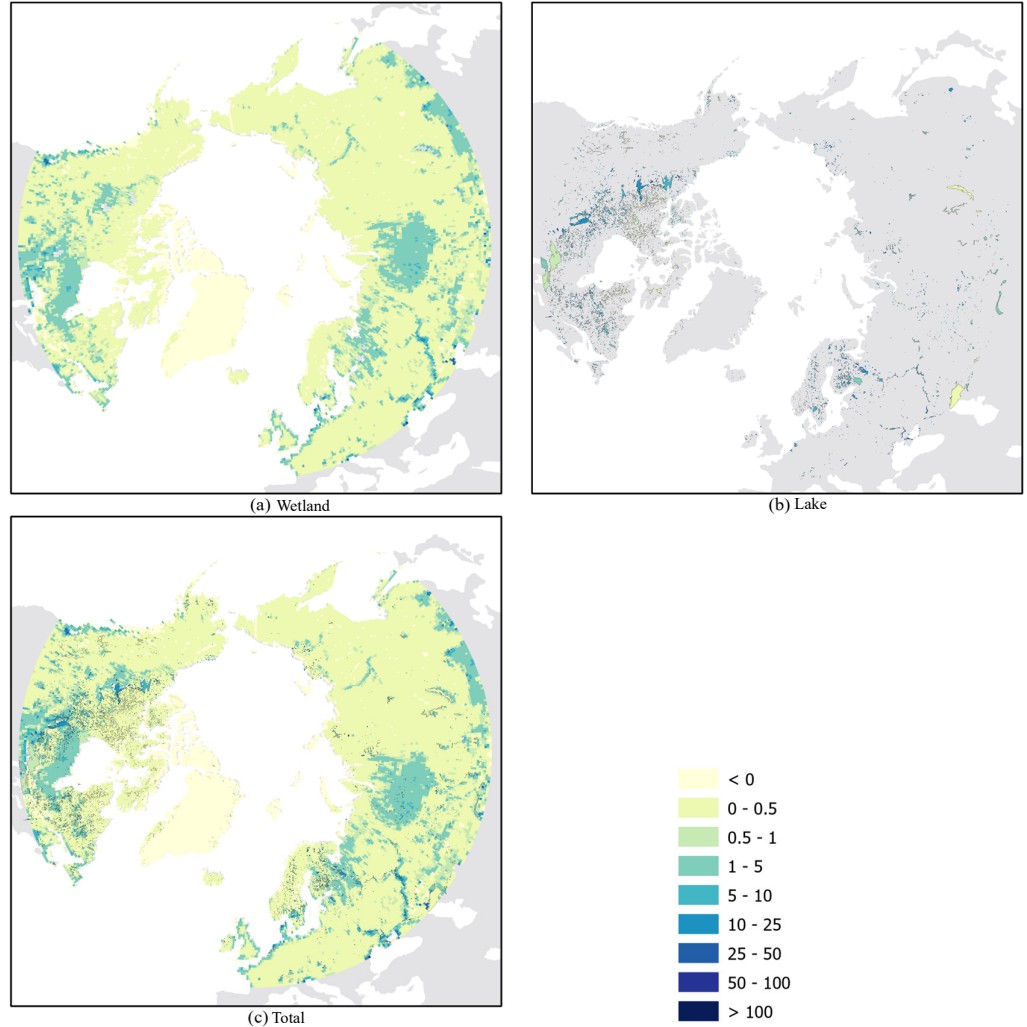

(a) Wetland

(b) Lake

(c) Total

| | |
|---|---|
| | < 0 |
| | 0 - 0.5 |
| | 0.5 - 1 |
| | 1 - 5 |
| | 5 - 10 |
| | 10 - 25 |
| | 25 - 50 |
| | 50 - 100 |
| | > 100 |

**Figure 2. Spatial distribution of average annual methane emissions (g CH$_4$ m$^{-2}$ yr$^{-1}$) from (a) wetlands, (b) lakes, and (c) total inland water systems in the pan-Arctic region.**

### 3.3. Correlation and sensitivity analysis results

The relationship between annual methane emissions from inland water systems and climate drivers as well as landscape areal change are shown in Fig. 3. The studied climate drivers include vapor pressure (relative humidity), precipitation, temperature, and shortwave radiation. For areal changes, wetlands and lakes are shown separately. We also did a correlation analysis between annual methane emissions and these drivers. The results are shown in Table 1. Temperature and vapor pressure have very similar trend and fit well with wetland emission with a high correlation of 0.80 and 0.88, which are the only two have the P-value less than 0.01. The precipitation captured the upward and downward trends of wetland

emissions, with a relatively high and statistically significant correlation of 0.56. Compared to which, the shortwave radiation and areal change have lower correlations with wetland emissions. For lake emissions, figure shows that temperature captured the most upward and downward trends, followed by shortwave radiation and precipitation with statistically significant correlations of 0.54, 0.47 and 0.45, respectively. Although the annual average vapor pressure shares a similar annual trend with temperature and lake methane emissions, the correlation analysis is relatively low. In addition, the methane emissions from lakes (0.56) are more sensitive to landscape areal changes than to wetlands changes (0.27 with no statistical significance).

**Table 1. Correlations between annual methane emissions and climate drivers and landscape changes**

|  | Shortwave Radiation | Precipitation | Temperature | Vapor Pressure | Areal Change |
|---|---|---|---|---|---|
| Wetland Emission | 0.20[d] | 0.56[b] | 0.80[a] | 0.88[a] | 0.35[d] |
| Lake Emission | 0.47[c] | 0.45[c] | 0.54[b] | 0.35[d] | 0.56[b] |

**(a) P-value less than 0.01; (b) P-value less than 0.05; (c) P-value less than 0.1; (d) P-value greater than 0.1**

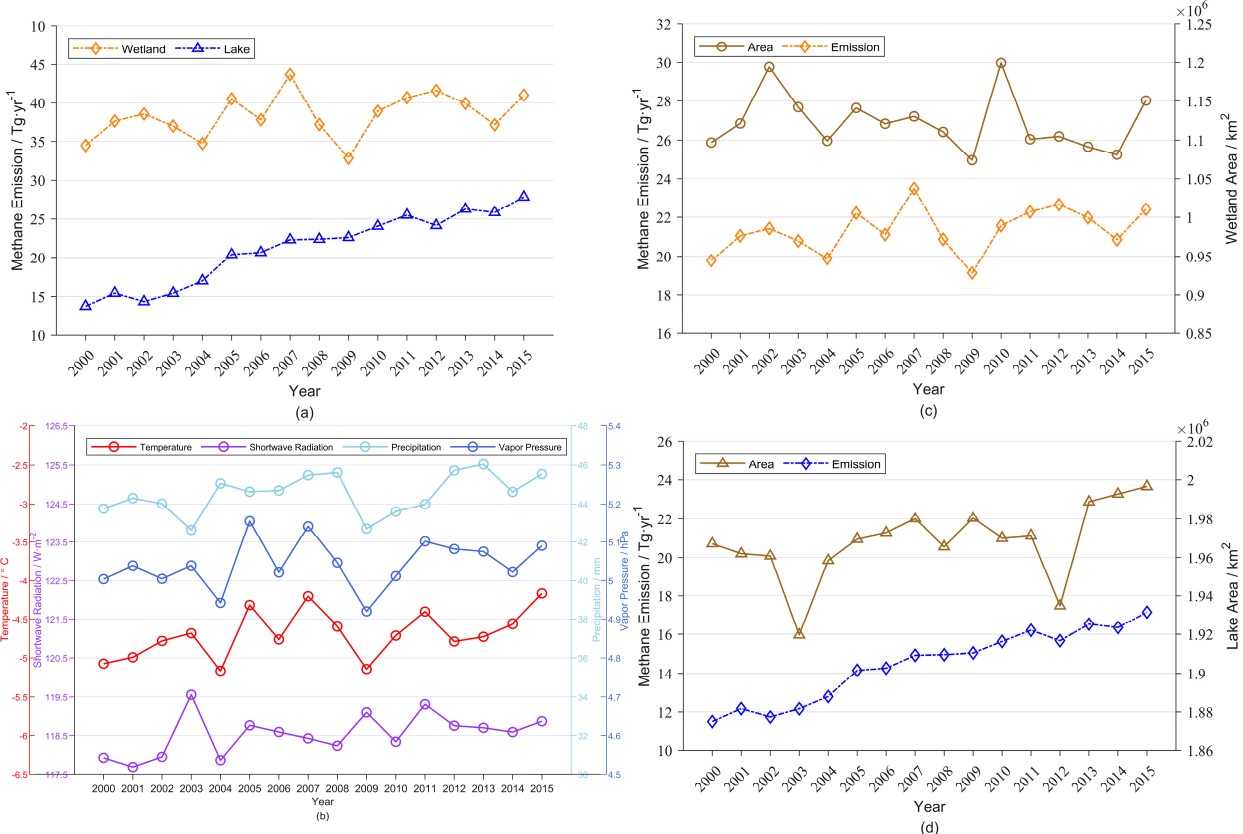

**Figure 3. Relationships between methane emissions from inland water systems and key drivers: (a) methane emissions, (b) 4 climate drivers, and methane emission with (c) wetland area, and (d) lake area changes.**

Considering that annual average values are not capable of capturing seasonal/monthly relationship, we then did another correlation analysis using monthly data. From which, monthly wetland and lake emissions, four climate drivers, and wetland inundation data were used while monthly lake area data are not available (Table 2). Each correlation in the table has a p-value lower than 0.01, which means they are all statistically significant. Vapor pressure, just like the high correlation with wetland emissions in

interannual trends (Table 1), the monthly correlation is still the highest among the five factors (0.96). The second highest correlation with wetland emissions is also temperature (0.89), followed by wetland area, shortwave radiation, and precipitation. Although the interannual variation of short-wave radiation not fully coincides with wetland emissions (Fig. 3) and they seem to have low and statistically meaningless correlation, their monthly correlation still has a relatively high value of 0.77. In terms of the correlation

of lake methane emissions, temperature has the highest value of 0.87, followed by relative humidity (vapor pressure) and precipitation. We also did a correlation analysis between wetland area and climate drivers and found that temperature and vapor pressure are the climatic factors that have the greatest impact on wetland landscape areal changes.

**Table 2. Correlation between monthly methane emission and climate drivers and landscape changes**

| | Shortwave Radiation | Precipitation | Temperature | Vapor Pressure | Areal Change |
|---|---|---|---|---|---|
| **Wetland Emission** | 0.77 | 0.76 | 0.89 | 0.96 | 0.79 |
| **Lake Emission** | 0.56 | 0.79 | 0.87 | 0.82 | |
| **Wetland Area** | 0.69 | 0.75 | 0.93 | 0.88 | |

Different types of lakes have various sensitivities to increasing temperature, precipitation, and additional lake sediment carbon (Fig. 4 and Table 3). Lake methane emission from above 45-degree north is more sensitive to temperature changes than to precipitation or lake sediment carbon pool. When temperature increases by 5° C, lake emissions increase by 19%, where thermokarst lakes are influenced the most (28.5%) and yedoma lakes are influenced the least (7.35%). Precipitation has low impacts on

lake $CH_4$ emissions. The overall lake emissions only increase by 0.19% when the precipitation increased by 15%. Thermokarst lakes remain relatively most sensitive to changes in precipitation (0.82), while the other three types of lakes were all insensitive. For additional sediment carbon added due to permafrost thaw, only thermokarst and yedoma lakes were impacted, with increasing by 15% carbon leading to a

similar increase for both types of lakes (20.85% and 18.98%), resulting in an overall CH$_4$ emission

increase by 6.85%.

**Table 3. Average increase for 4 types of lakes (temperate (TMP), boreal (BRL), thermokarst (TMK), and yedoma (YDM)) and total CH$_4$ emissions in 16-year period due to changes in temperature, precipitation, and lake sediment carbon.**

|  | TMP | BRL | TMK | YDM | Total |
| --- | --- | --- | --- | --- | --- |
| **Additional C** | 0 | 0 | 20.85% | 18.98% | 6.85% |
| **Temperature** | 19.24% | 22.38% | 28.49% | 7.35% | 18.81% |
| **Precipitation** | 0.12% | 0.05% | 0.82% | 0.06% | 0.19% |

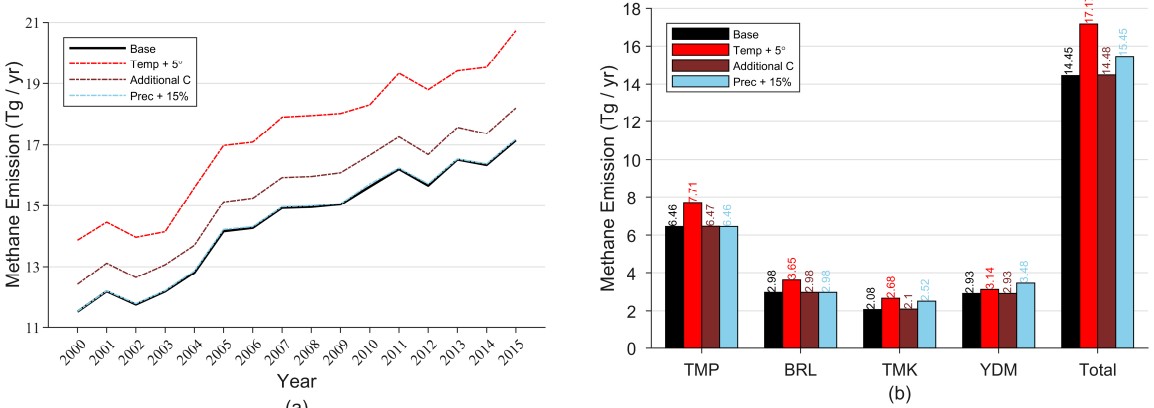

**Figure 4. Sensitivity test for increasing temperature by 5° C, increasing precipitation by 15%, and adding**
**additional 15% carbon into lake sediments (a); Average value of each type of lakes including temperate (TMP), boreal (BRL), thermokarst (TMK), and yedoma (YDM) (b).**

## 4. Discussion

### 4.1. Annual methane emissions from the landscape

From the previous studies, Wik et al. (2016) estimated 16.5 Tg CH$_4$ yr$^{-1}$ emissions from lakes and
ponds north of 50° N while Bastviken et al. (2011) estimated 13.4 for the inland waters (lakes, reservoirs, streams, and rivers) >54° N, both of which are estimated using measurement data combined with inventories. Based on a new spatially-explicit dataset of lakes > 50° N which includes not only all the lakes that area greater than 0.1 km$^2$ but also 6.5 million smaller lakes (0.02–0.1 km$^2$), Matthews et al. (2020) estimated the emissions are 13.8–17.7 Tg CH$_4$ yr$^{-1}$. Using a process-based model (bLake4Me, a
previous version of the ALBM model), Tan and Zhuang (2015a) estimated 11.86 Tg CH$_4$ yr$^{-1}$ in the year 2005-2008 ranging from 7.1 to 17.3 Tg CH$_4$ yr$^{-1}$ for north of 60° N. After this study, a coupled model of

bLake4Me and a thermokarst lake-evolution model was used to estimate a total methane emission of 11.3 ± 2.1 Tg $CH_4$ $yr^{-1}$ from lakes >60° N in the year 2006 (Tan and Zhuang, 2015b). Compared to these estimates, our lake simulation results fall in a reasonable range.

For emissions from northern high latitude wetlands, Chen et al. (2015) estimated 36.1 ± 6.7 Tg $CH_4$ $yr^{-1}$ during 1997–2006 for the same pan-Arctic wetlands (north of 45° N) using an enhanced Variable Infiltration Capacity (VIC) model linked with the Walter and Heimann wetland $CH_4$ emissions model. Zhang et al. (2017) used a bottom-up approach with LPJ-wsl model, estimating methane emissions of 23.4 ± 0.76 Tg $CH_4$ $yr^{-1}$ from wetlands > 50° N over the period 1980-2000. Poulter et al. (2017) used an ensemble of biogeochemical models constrained with remote sensing surface inundation and inventory-based wetland area data (SWAMPS‑GLWD, a previous version of WAD2M used in this study) estimating the boreal wetland emitted 44 ± 19 Tg $CH_4$ $yr^{-1}$ in 2012. Using TEM-MDM, but combined with different transient wetland inundation area fraction datasets, Liu et al. (2020) estimated the emissions are 38.90 Tg $CH_4$ $yr^{-1}$ from the region 45-90° N.  Our estimates are at the lower end of these records. We attribute this to the change in inundation area data. The larger lake extent in GRWL & HydroLAKES compared to GSW dataset leads to downward-revised wetland area in WAD2M Version 2.0 versus Version 1.0. The revision in version 2.0 slightly reduced vegetated wetland extent in the mid-latitudes especially for the region 45-70° N, which is the portion with the most methane emissions in our study area. That could explain the gap between our results and the previous ones. In addition, compared to other model simulations that were also involved in the same project (Global Carbon Project wetland $CH_4$, GCP-$CH_4$) where 16 models give an annual average $CH_4$ emission of 28.8 ± 11.8 Tg $CH_4$ $yr^{-1}$ from northern wetlands >45° N in 2000–2020, our simulation result of 21.69 ± 0.59 $CH_4$ Tg $yr^{-1}$ lays in a reasonable range.

       Besides biogeochemistry modeling approaches, atmospheric chemistry transport and inversion models have also been used to constrain the methane emission quantification from pan-Arctic wetlands and lakes. Bruhwiler et al. (2014) developed an assimilation system for atmospheric $CH_4$ and simulated the annual emissions from the wetland over the northern high latitudes (53–90° N) of about 23 Tg $CH_4$ $yr^{-1}$. Tan et al. (2016) used a nested-grid high-resolution inverse model estimating methane emissions from north of 60° N in the range of 11.9–28.5 Tg $CH_4$ $yr^{-1}$, of which wetlands and lakes accounted for 5.5–14.2 and 2.4–14.2 Tg $CH_4$ $yr^{-1}$, respectively.

**Table 4. Comparison with previous studies with different methods. The average values are estimated by weighting the area.**

| Method | Type | Reference | Study Area | Emissions (Tg $CH_4$ yr$^{-1}$) | Average | Our result |
|---|---|---|---|---|---|---|
| **Bottom-up** | Wetland | Chen et al. (2015) | > 45° N | 36.1 ± 6.7 | 36.3 | 21.69 |
| | | Zhang et al. (2017) | > 50° N | 23.4 ± 0.76 | | |
| | | Poulter et al. (2017) | Boreal region | 44 ± 19 | | |
| | | Liu et al. (2020) | > 45° N | 38.9 | | |
| | Lake | Bastviken et al. (2011) | > 54° N | 13.4 | 17.6 | 14.76 |
| | | Wik et al. (2016) | > 50° N | 16.5 | | |
| | | Tan and Zhuang (2015a) | > 60° N | 11.86 | | |
| | | Tan and Zhuang (2015b) | > 60° N | 11.3 ± 2.1 | | |
| | | Matthews et al. (2020) | > 50° N | 13.8–17.7 | | |
| **Top-down** | Wetland and Lake | Tan et al. (2016) | > 60° N | 11.9–28.5 | 29 | |
| | | Bruhwiler et al. (2014) | > 53° N | 23 | | |

Our simulation shows that methane emissions from inland water systems in the pan-Arctic are 36.46 ± 1.02 Tg $CH_4$ yr$^{-1}$, which are in the middle of bottom-up estimates of 53.9 Tg $CH_4$ yr$^{-1}$ and top-down estimates of 29 Tg $CH_4$ yr$^{-1}$ from previous studies (Table 4). Our bottom-up model estimates are much lower than the previous bottom-up estimates and closer to the previous top-down estimates. We attribute this to using two non-overlap dynamic areal change datasets to minimize the "double accounting" problem raised by Thornton et al. (2016).

## 4.2. Climate drivers and sensitivity analysis

Since the climate variables often co-vary over time, some of them could be confounders during the correlation analysis. Thus, the correlation analysis may not reflect the 'true' sensitivity of methane fluxes to single climate variable (Table 2). A partial correlation analysis is then conducted to eliminate the covariate effects between climate drivers for better analyzing correlation between each individual variable and methane emissions. We first noticed that interannual average vapor pressure and temperature have a relatively high correlation (Fig. 3b) with a value of 0.84 and 0.96 (both P-value are much less than 0.01) for annual and monthly data, respectively. This is to be expected since the vapor pressure is greatly affected by temperature and even itself is calculated from temperature and relative humidity data. We also found, while the correlation between annual temperature and radiation is not strong (0.40 with p-

value over 0.1), the correlation is strong at monthly time step (0.81 with P-value much less than 0.01), this may explain the relatively high correlation between monthly shortwave radiation and methane emissions. In addition, precipitation is greatly affected by vapor pressure and temperature, and there is a strong correlation among them (0.83 for vapor pressure and 0.77 for temperature). Hence our partial correlation analysis aims to examine the interannual and seasonal relationship between methane emissions and (1) vapor pressure and shortwave radiation after removing the thermal effect of temperature (Vapr/T and SwRd/T), (2) temperature independent of radiation (Temp/R), and (3) precipitation eliminating the impact of temperature and vapor pressure (Prec/TV) (Table 5).

**Table 5. Partial correlations for shortwave radiation eliminating temperature (SwRd/T), vapor pressure independent of temperature (Vapr/T), temperature independent of radiation (Temp/R), and precipitation eliminating temperature and vapor pressure (Prec/TV).**

| Time scale | Tpye | SwRd/T | Prec/TV | Temp/R | Vapr/T |
|---|---|---|---|---|---|
| Seasonal | Wetland | 0.20[a] | -0.54[a] | 0.72[a] | 0.86[a] |
| | Lake | -0.47[a] | 0.56[a] | 0.85[a] | -0.14[c] |
| Annual | Wetland | -0.23[d] | 0.35[d] | 0.81[a] | 0.65[a] |
| | Lake | 0.33[d] | 0.43[d] | 0.45[d] | -0.23[d] |

(a) p-value less than 0.01; (b) p-value less than 0.05; (c) p-value less than 0.1; (d) p-value greater than 0.1

Although temperature and vapor pressure still are the most important drivers to the annual seasonal wetland methane emissions, vapor pressure independent of the thermal effect are no longer the main driver of lake methane emissions. For seasonal wetland emissions, vapor pressure has the highest coefficient and temperature has the second highest ones, consistent with our previous correlation analysis, high vapor pressure may limit the stomatal opening and reduce evapotranspiration, thus increases soil moisture which could stimulate methane production (Zhuang et al., 2003). Other studies also indicated that the impact of wet/dry cycles on regional methane emissions is evident (e.g., Watts et al., 2014). When it comes to annual trend, temperature tends to have higher influence on wetland methane emissions, indicating that wetland is more sensitive to temperature in a long term than vapor pressure. For lake emissions, vapor pressure has less impact when eliminating the temperature, showing that their high correlation is mostly induced by the thermal effect. In our model, lake methane emission is mainly through two processes, methane ebullition and diffusion (Tan et al., 2015, 2017). High vapor pressure would suppress water methane diffusion, but the influence is relatively small compared to overall

emissions. Shortwave radiation with temperature effect removed has a much smaller effect on seasonal emissions, indicating the high correlation of radiation is caused by the heating effect of radiation and the high sensitivity of temperature in our model. We believe the results of partial correlation analysis capture the relationship between inland water systems methane emissions and climate drivers.

The sensitivity analysis suggests that a 5 °C increase in temperature increases the pan-Arctic lake methane emission by 20%. Compared to previous studies, Guo et al. (2020) estimated a 40% lake methane emission increase for the same study area by the end of the 21$^{st}$ century in the scenario that the temperature increases around 7.5 °C. Sepulveda-Jauregui et al. (2018) showed that for subarctic oligotrophic lakes, increasing lake water temperature by 2 °C leads to a net increase in $CH_4$ emissions by 47-56%. However, their work did not consider the ice cover season of high-latitude lakes, from which the methane fluxes can be blocked by a thick layer of ice for several months each year and then oxidized in the water column. In addition, the relatively low response of yedoma lakes (~7%) to the increasing temperature could be explained by their mobilized labile carbon is usually in deep sediments (Tan and Zhuang, 2015a), which means that the influence of the warming air temperature will take much longer to enhance methane production in the lake sediment. In contrast, when we directly increase the labile carbon density ($C_{labile}$) at the water-sediment interface, the methane emission of yedoma lakes increased much higher (~19%), while the thermokarst lake were affected less (~20%) compared to its response to temperature change (~28%). For precipitation, although it was set in the model to bring the load of allochthonous carbon to the lake (Tan et al., 2017), increasing it by 15% only makes a negligible impact on methane emission. A plausible explanation is that the lakes are relatively saturated with extraneous carbon in sediments, so any increase brought by the additional precipitation tends to have small influences.

### 4.3. Uncertainty analysis and future works

Although our simulation results more accurately estimate methane emissions from inland water systems in the pan-Arctic by avoiding the "double accounting" problem, there still exist some uncertainty sources in this study. First, despite the use of two non-overlapping landscape change maps to avoid the uncertainty caused by "double accounting", the precision of the two maps remains to be examined. The HydroLAKES database used in the GLCP and WAD2M datasets only contains lakes and reservoirs which area greater than 0.1 km$^2$ (Messager et al., 2016), which means that lakes and ponds smaller than 0.1 km$^2$

are either not considered or misclassified as wetlands. Those small lakes and ponds cover in total about

$1 \times 10^6$ km$^2$ which equals more than half the area of Alaska (Verpoorter et al., 2014). Also, some previous studies have found higher methane fluxes in small and shallow lakes (Holgerson et al., 2016; Sasaki et al., 2016), and lakes appear to emit more methane than wetlands, implying that lake methane emissions may still be underestimated. Secondly, during the simulation, although we classified the lakes based on their sediment type, size, and depth, we still assumed that all the same types of lakes to be homogeneous

which were assigned to the same set of parameters. Nevertheless, lakes are highly heterogeneous across the globe (Guo et al., 2021), especially for those big lakes, such that regional lake simulation may introduce a high uncertainty.

Furthermore, recent studies have found that groundwater discharge could be an important pathway as lateral $CH_4$ inputs to Arctic lakes that links $CH_4$ production in thawing permafrost to atmospheric

emissions via lakes (Olid et al., 2022). Jammet et al. (2015) also confirmed that spring is a crucial period for methane dynamics in subarctic shallow lakes while large methane emissions were observed during the spring thaw. These two important processes were not considered in our process-based ALBM model. Similarly, compared to other model results in GCP-$CH_4$ projects, our TEM-MDM modeled wetland $CH_4$ emissions are relatively low in subzero temperature months, while a field study found that substantial

emissions occur during the "zero curtain" period, when subsurface soil temperatures are poised near 0 °C (Zona et al., 2016). Therefore, our next step will be modifying the TEM-MDM and ALBM models by taking those important processes into consideration. In addition, higher resolution maps of dynamic wetland inundation and lake landscape changes are highly needed.

## 5. Conclusions

By using two dynamic areal change datasets combined with process-based terrestrial and lake biogeochemical models, we are among the first to quantify methane emissions from both land and aquatic inland water systems, i.e., wetlands and freshwater bodies in the pan-Arctic, which avoids the uncertainty caused by area "double accounting". Our simulations indicate that the total methane emissions from pan-Arctic inland water system are $36.46 \pm 1.02$ Tg $CH_4$ yr$^{-1}$ during 2000-2015, of which wetlands and lakes

were $21.69 \pm 0.59$ Tg yr$^{-1}$ and $14.76 \pm 0.44$ Tg yr$^{-1}$, respectively. Our estimation narrows the difference between previous estimates using 'bottom-up' and 'top-down' methods. In the pan-Arctic, wetland

methane emissions are most affected by vapor pressure, followed by temperature, while lake emissions are more sensitive to temperature than to precipitation and landscape areal change. Furthermore, the methane emissions from lakes are more sensitive to annual landscape areal changes than from wetlands.

West Siberia Lowland and the Hudson Bay Lowland were the two strong sources of wetlands and lakes have higher emissions around Mackenzie River delta of Canada and the Hudson Bay Lowland area. In addition, lakes emit more methane than wetlands under the same condition. Although the lack of understanding of the underlying methane cycle mechanisms in the lake makes the response of $CH_4$ emissions from Arctic lakes to climate change highly uncertain, our sensitivity test using the process-

based model ALBM does indicate the pan-Arctic Lake $CH_4$ emissions are influenced by increasing temperature more compared to lake sediment carbon increase.

*Code and data availability.* The data used to reproduce figures, codes, model and samples of running directory can be accessed via Purdue University Research Repository:

https://purr.purdue.edu/publications/4166/1.

*Author contributions.* QZ designed the study. XL conducted model simulation and analysis. XL and QZ wrote the paper.

*Competing interests.* The authors declare that they have no conflict of interest.

*Acknowledgments.* This study is supported through a projected funded to Qianlai Zhuang by NASA (NNX17AK20G) and a project from the United States Geological Survey (G17AC00276). We acknowledge the Rosen High Performance Computing Center at Purdue for computing support. We also

acknowledge the World Climate Research Programme's Working Group on Coupled Modeling Intercomparison Project CMIP5, and we thank the climate modeling groups for producing and making available their model output.

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
