# Peer review of "Methane emissions from Arctic landscapes during 2000-2015: An analysis with land and lake biogeochemistry models"

_Biogeosciences, 2022_

## Author Response (AR1)

**Response for manuscript "Methane emissions from Arctic landscapes during 2000-2015: An analysis with land and lake biogeochemistry models":**

**Response to comment #1**

Dear Anonymous Referee #1:

We sincerely thank you for the valuable comments that help improve the quality of our manuscript. Following your suggestions, we have revised the manuscript. The comments are laid out below in bold italic font and specific concerns have been numbered. Our response is given in normal font.

*Specific comments:*

1. *Line 16-17: What are the uncertainties?*

   We have rerun the simulation with a different input of 20CRv3-ERA5 datasets to take the uncertainty of input data into consideration. We then rewrote the results and reproduced figure 1(a) to add error bars.

2. *Line 20-22: As previously mentioned the emissions are significantly affected by humidity and vapor pressure.*

   Based on the partial correlation analysis, when eliminating the temperature effect, we found that vapor pressure still has the highest influence on wetland methane emissions while has little impact on lake emissions. We rewrote this paragraph with our new result and interpretation in this revision.

3. *Line 27: IPCC 2014 is not the latest one. IPCC6 is published one year ago. What is the newest IPCC6 report number?*

   We have looked through IPCC AR6 report and updated the number. The newest number is around 28 times higher than $CO_2$ (27.2 in non-fossil origin and 29.8 in fossil origin).

4. *Line 29-35: I feel people will get confused about these introductions about wetland emissions. It would be more clear if put them together instead of two paragraphs.*

   We have combined the two paragraphs together and reconstructed them to avoid confusion.

5. *Line 40-42: People may expect a little bit more mechanisms of lake emission. Like why it will be high in spring?*

   We have checked the literature and added more details about lake emission mechanisms to explain its high emission during spring ice melt.

6. *Line 59-61: 1) As you mentioned the resolution will be the key issue for the double counting issue. Then what is the resolution for the data you used or what model resolution should be highlighted here. 2) Or if*

*resolution is not the matter. You may need want to mention why involving those two datasets or models can avoid double counting.*

Static wetland and lake landscape maps may always have some overlaps irrespective of their resolutions, coarser resolutions may lead to larger overlaps. In our study, we used two dynamic area datasets for wetlands and lakes, which minimize the potential overlap between water bodies and wetlands.. We have added the explanation in the text to discuss how these datasets avoid overlap.

7. *Line 60: "unpublished data". Is the data going to be published somewhere? How people are going to access them*

The dataset is in prepared for publication. We have also cited AGU abstract as a reference

8. *Line 100-105: What are the input of vegetation types, soil types coming from?*

We have added the references for the data of vegetation and soil types. Vegetation type distribution are from Melillo et al. (1993) and soil texture are from Zhuang et al. (2003).

9. *Line 118-120: This part needs to be highlighted and extended for details.*

We have added details to explain why using these two datasets could minimize "double accounting".

10. *Line 135: Here should be "wind speed at 10 m"*

We corrected it.

11. *Line 137-138: Are they the same as what are inputted in ALBM*

They are the same. We have mentioned in our manuscript: "These forcing data were used for both models to ensure that no additional uncertainties are introduced."

12. *Line 154: Should be "(using TEM-MDM)"*

This part has been corrected.

13. *Line 158-159: Have you mentioned what data you used for spin-up?*

We didn't mention the detail of spin-up data, we have added them in this revision.

14. *Line 167: Under what scenarios? and we have the latest IPCC6 report, have you checked whether the number is changed?*

We have looked through the IPCC AR6 report and other reference. Under SSP5-8.5 scenario, the temperature will increase roughly by 4-6 °C and the precipitation exhibits an increasing trend at a rate of 10.28 mm/decade in the northern hemisphere, corresponding to ~13-18% increase by the end of the 21st century.

15. *Line 174-176: Why do you suddenly mention the isotope here? I don't think you have used any of the isotope-related analyses.*

We have changed wording to "old organic carbon pool".

16. *Line 182-183: What is the uncertainty of your simulation in each year? Some analyses like considering the uncertainty of input data may be good to be included.*

Like our response to comment #1, we have rerun the simulation with a different input of 20CRv3-ERA5 datasets to take the uncertainty of input data into consideration.

17. *Line 184: How are those uncertainties coming from?*

The previous numbers are mean and standard deviation. We have changed them to include the input uncertainty.

18. *Figure 1 and table 1: Fig1a is duplicated with table 1. Also, table 1 is more suitable for a time-series plot*

We have deleted the redundant table 1 and left figure 1.

19. *Line 209-211: I don't understand here.*

We have rewritten this part.

20. *Figure 2: Please label the simulation type also in the figure instead of only the caption.*

We have updated the figure with caption.

21. *Table 2: why do some of the results have no superscript? Also, these superscripts should be explained in a footnote instead of the table title.*

The results without superscript mean that their P-value greater than 0.1. We have added it in the footnote based on your suggestion.

22. *Table 3: why 0 values here?*

We have mentioned it in the method part. "**Because the old organic carbon pool may contribute to $CH_4$ production in the permafrost thaw bulb under yedoma and thermokarst lakes, we just altered the corresponding $C_{labile}$.**" Therefore, for temperate (TMP) and boreal (BRL) lakes which do not have permafrost under the sediment, they will not be influenced by permafrost thaw.

23. *Figure 4: So can I understand this way that the high correlation of radiation is caused by the heating effect of radiation and the high sensitivity of temperature in your model?*

In this revision, we conducted partial correlation analysis. Now the correlation between shortwave radiation and emissions is not significantly high.

24. *Line 264: I didn't see how your simulation results fit in the previous ranges. Could you summarize the results and also your results for comparison?*

We have added a new table to summarize the results for comparison based on your suggestion.

25. *Line 275: This part is interesting and should be talked about more about why you can attribute this.*

We have added more detail about the reason. The larger lake extent in GRWL & HydroLAKES compared to GSW dataset leads to downward-revised wetland area in WAD2M Version 2.0 versus Version 1.0. The revision in version 2.0 slightly reduced vegetated wetland extent in the mid-latitudes especially for the region 45-70° N, which is the portion with the most methane emissions in our study area. That could rationally explain the gap between our results and the previous estimates.

26. *Line 288-290: This is a very interesting conclusion since we always know that there are huge differences between bottom-up modeling and top-down modeling results of CH4 emission. I may expect to see emphasizing more explicitly how the uncertainty is reduced based on your study and what ranges you suggested.*

We reviewed literature and developed a new Table (Table 4) to help address this comment.   We added the following to main text "*Our simulation shows that methane emissions from inland water systems in the pan-Arctic are $36.46 \pm 1.02$ Tg CH$_4$ yr$^{-1}$, which are in the middle of bottom-up estimates of 53.9 Tg CH$_4$ yr$^{-1}$ and top-down estimates of 29 Tg CH$_4$ yr$^{-1}$ from previous studies (Table 4).   Our bottom-up model estimates are much lower than the previous bottom-up estimates and closer to the previous top-down estimates.   We attribute this to using two non-overlap dynamic areal change datasets to minimize the "double accounting" problem raised by Thornton et al. (2016).*"

27. *Line 296-298: Are they possibly due to your study area being mostly located in boreal so the seasonal cycle of all inputs and methane emissions are strong? Then they should be easily correlated. Their being highly correlated may not be simply explained as they are significant to wetland emission, since another possibility is that they can be confounders, correlated with each other (e,g, SR, and temperature). Causal relations may be considered here*

Based on partial correlation analysis and removing the covariance among the climate drivers (shortwave radiation eliminating temperature, vapor pressure independent of temperature, temperature independent of radiation, and precipitation eliminating temperature and vapor pressure), we have revised Discussion 4.2 to address the comments.

Dear Anonymous Referee #2:

Thanks for the valuable comments that help improve the quality of our manuscript. Below we detailed how we responded to your comments. The comments below are in bold italic font and specific concerns have been numbered. Our responses are given in normal font.

*Main questions*

***1.   Since reducing the uncertainties from 'double accounting' is one of the main focus for this manuscript, I wonder how large the amount was 'double accounted' and how large the estimate was improved by your approach. Could you elaborate in the abstract and main text?***

In this revision, we summarized the previous 'bottom-up' and 'top-down' results and estimated the average emissions, the 'bottom-up' estimates are 53.9 Tg $CH_4$ $yr^{-1}$ and "top-down" estimates are 29 Tg $CH_4$ $yr^{-1}$). The estimate of the pan-Arctic $CH_4$ budget shown in Thornton et al. (2016) is 59.7 Tg $yr^{-1}$, ranging 36.9–89.4 Tg $yr^{-1}$ for 'bottom-up' and 23 ± 5 Tg $yr^{-1}$ for 'top-down'. Thus, our results of **36.46 ± 1.02 Tg $CH_4$ $yr^{-1}$** narrow those earlier estimates. We have added discussion into Abstract and main text.

***2.   The Lake with a size < 10 ha is the central component that affects estimating wetland and lake CH4 emissions simultaneously. It is unclear how you treat this since the HydroLakes doesn't cover the extent for small lakes?***

Like we discussed in Section 4.3, HydroLAKES database used in the GLCP and WAD2M datasets only contains lakes and reservoirs that have area greater than 0.1 $km^2$, suggesting that lakes and ponds smaller than 0.1 $km^2$ in our study are either not considered or misclassified as wetlands. Thus, we admit that our estimates have not completely eliminated the double accounting, rather we might have reduced it to some extent as discussed in our response to comment #1.

***3.   The statistical analysis needs to be improved. I am surprised to see shortwave radiation has a much higher correlation with CH4 emissions than the temperature for the Arctic wetlands and lakes on the annual basis. What's the underlying mechanism for this? Also, these climate variables often co-vary over time. the analysis done in its current way may not reflect the 'true' sensitivity of CH4 fluxes to the climate variables.***

In this revision, we conducted partial correlation analysis to remove the covariance among the climate drivers (shortwave radiation eliminating temperature, vapor pressure independent of temperature, temperature independent of radiation, and precipitation eliminating temperature and vapor pressure). We have revised Discussion 4.2 with correlation analysis.

***4.   The description of the sensitivity test is not clear. How did you treat the increasing temperature by 5 deg C and increasing precipitation by 15%? Was the increased temperature or precipitation evenly allocated to each month or proportional to its seasonal cycle? This is important because the added precipitation and temperature to different seasons would have different effects on CH4 emissions. Also, how the threshold for temperature and precipitation were chosen.***

Specifically, we uniformly increased 5 deg C for daily temperature and 15% for daily precipitation, respectively. We have added details to describe how we choose the threshold. "**According to the previous studies, under the SSP5-8.5 scenario, the temperature will increase roughly by 4-6 °C (IPCC, 2021; Huang et al., 2022) and the precipitation exhibits an increasing trend at a rate of 10.28 mm/decade in the northern hemisphere, corresponding to ~13-18% increase by the end of the 21st century (Chen et al., 2014; Du et al., 2022).**"

*Specific comments:*

1. *Table 2. The way its presented is confusing. As I said, your might want to consider a multiple regression or partial correlation to analyze the sensitivity.*

We added partial correlation analysis result and discussion. We have reconstructed the discussion 4.2 with correlation analysis.

2. *Figure 3. It is difficult to read as it is currently presented. Could you minimize the duplicated information by plotting the time series by different groups? Say Ch4 emissions and climate variables as two groups?*

We reproduced Figure 3 by putting emission and climate driver separately.

3. *Discussion 4.1 it is not clear to me how your number narrows down the double counting.*

See our response to main comment #1.

4. *4.2 is confusing. If analyzing yearly values can not capture the 'true' relationship as the authors said. Why present it in results? To me it looks like you are analyzing the driving variables for different time scales, The 4.2 is for seasonal cycle and the annual analysis is for interannual variations.*

We presented monthly correlation results together with annual results. In addition, we also did partial correlation analysis to eliminate the covariate effect and revised Discussion 4.2.

5. *Section 4.2 Why vapor pressure has the highest correlation for monthly results. This needs an explanation for the mechanisms of dominant control by humidity and vapor pressure. Are the mechanisms differ with wetlands and lakes?*

Based on the partial correlation analysis, we found that vapor pressure still has the highest influence on wetland methane emissions while has little impact on lake emissions. We revised Section 4.2 based on partial correlation analysis.

6. *Line 304 Correlation of 0.77 for short-wave radiation is still high.*

Partial correlation analysis shows the correlation is not high in this revision.

Sincerely,

The Authers.